 

# A SLC4 family bicarbonate transporter is critical for intracellular pH regulation and biomineralization in sea urchin embryos

Marian Y Hu[1]*, Jia-Jiun Yan[1,2], Inga Petersen[1], Nina Himmerkus[1], Markus Bleich[1], Meike Stumpp[3]

[1]Institute of Physiology, Christian-Albrechts University of Kiel, Kiel, Germany; [2]Institute of Cellular and Organismic Biology, Taipei, Taiwan; [3]Comparative Immunobiology, Institute of Zoology, Christian-Albrechts University of Kiel, Kiel, Germany

**Abstract** Efficient pH regulation is a fundamental requisite of all calcifying systems in animals and plants but with the underlying pH regulatory mechanisms remaining largely unknown. Using the sea urchin larva, this work identified the SLC4 $HCO_3^-$ transporter family member *SpSlc4a10* to be critically involved in the formation of an elaborate calcitic endoskeleton. *SpSlc4a10* is specifically expressed by calcifying primary mesenchyme cells with peak expression during de novo formation of the skeleton. Knock-down of *SpSlc4a10* led to pH regulatory defects accompanied by decreased calcification rates and skeleton deformations. Reductions in seawater pH, resembling ocean acidification scenarios, led to an increase in *SpSlc4a10* expression suggesting a compensatory mechanism in place to maintain calcification rates. We propose a first pH regulatory and $HCO_3^-$ concentrating mechanism that is fundamentally linked to the biological precipitation of $CaCO_3$. This knowledge will help understanding biomineralization strategies in animals and their interaction with a changing environment.

DOI: https://doi.org/10.7554/eLife.36600.001

*For correspondence: m.hu@physiologie.uni-kiel.de

Competing interests: The authors declare that no competing interests exist.

## Introduction

Sea urchin larvae with their elaborate calcareous endoskeleton have been studied by embryologists for over a century to promote our understanding of calcification in biological systems (*Boveri, 1901*; *Decker and Lennarz, 1988*; *Wilt, 2002*). Similar to mammalian osteoblasts that arise from mesenchymal stem cells (MSC), the sea urchin larval skeleton is also produced by a specific cell line – the primary mesenchyme cells (PMCs) (*Wilt, 2002*). During the blastula stage, PMCs ingress into the blastocoel and migrate in a stereotypical pattern, forming a posterior ring around the blastopore (*Ettensohn, 1990*; *Gustafson and Kinnander, 1956*). Amorphous calcium carbonate (ACC) is precipitated within intracellular vesicles that exocytose their content into the lumen of a syncytial cable formed by the PMCs. This process involves at least 40 distinct skeletal matrix proteins supporting the formation of the mature calcite spicules within this extracellular space (*Beniash et al., 1997*; *Benson et al., 1986*). Some of these matrix proteins are also present in intracellular compartments where they may play a role in the stabilization of ACC (*Urry et al., 2000*; *Wilt, 2002*). Furthermore, recent findings suggest that $Ca^{2+}$ also enters calcification vesicles by endocytosis of seawater into a vesicular network within the PMCs (*Vidavsky et al., 2016*).

During de novo formation of the larval skeleton in the mid-gastrula stage, calcium uptake increases ten-fold, compared to the blastula stage. 70% of this calcium is incorporated into the newly formed spicules (*Nakano et al., 1963*), while the remaining 30% reflects the uptake of calcium into cell organelles, including mitochondria and the smooth endoplasmic reticulum (*Decker et al.,*

**eLife digest** Many marine organisms such as mussels, sea urchins or corals, have skeletons and shells, which – due to their beautiful colors and shapes – are often desirable collector pieces. These structures are made from calcium and carbonate ions that react to form calcium carbonate crystals in a process known as biomineralization.

In sea urchin larvae, for example, the skeleton is built by so-called primary mesenchyme cells, which work similar to the bone forming cells in mammals. These mesenchyme cells use calcium from the sea water, which travels to the site where the shell starts to form. About half of the carbonate comes from carbon dioxide that the animals make as they breathe, but it is not known how the other half gets to the site of biomineralization.

Producing a skeleton generates acid, and marine animals need to be able to regulate their pH levels, as too acidic environments can dissolve the calcium carbonate and threatening to destroy the developing shell. How cells accumulate enough carbonate to make their shells, and how they cope with acidity is still poorly understood.

Here, Hu et al. address this problem by studying purple sea urchin larvae, revealing that they use ion transporters to gather bicarbonate from seawater. These structures are part of a group of bicarbonate transporters known as the 'SLC4 transporter family', which sit across the membrane of the mesenchyme cells and move the bicarbonate ions along. As the sea urchin larvae develop, the levels of the transporter protein start to rise in mesenchyme cells, peaking around the time they are producing the skeleton.

Stopping the production of the transporter hindered the larvae from building normal skeletons and also made their cells more acidic. It turns out that bicarbonate is more than a skeleton ingredient – it also helps to buffer the acid made in the process. Bicarbonate ions can combine with acidic molecules to form water and carbon dioxide. Bicarbonate pumped in from the sea neutralises the acidic molecules made during calcium carbonate formation, which helps to stabilize pH levels.

When the acidity of the water was experimentally increased, it prompted the sea urchins to produce more of the SLC4 transporters, revealing that they may have another role to play. Their acid-neutralizing capability helped the animals to cope with changes in their environment. Taking on more bicarbonate could therefore help to compensate for rising acidity, allowing skeleton production to carry on as normal.

This last finding is important in the context of ocean acidification. As the amount of carbon dioxide in the atmosphere increases, more of the gas dissolves in the sea. The chemical reactions that follow make the water more acidic and decreases the pH levels of the sea. Understanding how animals make their skeletons and shells, and manage acid, could reveal how they will cope as the environment changes in the future.

DOI: https://doi.org/10.7554/eLife.36600.002

*1987*). Besides the availability of $Ca^{2+}$, dissolved inorganic carbon (DIC, i.e. $CO_2$, $HCO_3^-$ and $CO_3^{2-}$) is required at equimolar levels for the precipitation of $CaCO_3$. Radioisotopic measurements demonstrate that up to 63% of carbon for spicule formation derives from respiratory $CO_2$, while the remaining DIC that is incorporated into the spicules derives from seawater (*Sikes et al., 1981*). Based on the assumption that $HCO_3^-$ rather than $CO_2$ or $CO_3^{2-}$ supplies the remaining 37% of DIC for calcification, 1.6 protons are generated per molecule $CO_3^{2-}$ precipitated. An even stronger local acidification at the site of ACC formation can be expected considering $CO_2$ as the major source of DIC. Based on these calculations, it has been suggested that PMCs are capable of efficiently handling this proton load using pH regulatory mechanisms (*Sikes et al., 1981*). However, such potential acid-base regulatory mechanisms in PMCs remain poorly understood despite their importance in the formation of the larval skeleton.

A previous study demonstrated that $pH_i$ regulation of PMCs is dependent on external $Na^+$ as well as $HCO_3^-$ to compensate for an intracellular acidosis indicating $Na^+$-dependent $HCO_3^-$ buffering mechanisms (*Stumpp et al., 2012*). The importance of $HCO_3^-$ transport in the calcifying PMCs is underlined by the observation that DIDS (4,4′-diisothiocyano-2,2″-disulfonic acid stilbene), an inhibitor for most SLC4 family transporters (*Romero et al., 2004*) inhibits uptake and deposition of $^{45}Ca$

into spicules (*Yasumasu et al., 1985*). Accordingly, it has been concluded that these compounds block the supply of $HCO_3^-$ for ACC formation rather than the ability of PMC cells to form their syncytium as shown for other blockers (*Basse et al., 2015*; *Mitsunaga et al., 1986*). Furthermore, $H^+$-export pathways involving $Na^+/H^+$ exchange were suggested to remove protons from the cytosol of PMCs although amiloride, an inhibitor for $Na^+$-dependent $H^+$ exchange had no effect on spicule formation and $pH_i$ regulation of PMCs (*Mitsunaga et al., 1987*; *Stumpp et al., 2012*). Here, we hypothesize that $HCO_3^-$ transport pathways, *via* a so far unidentified $HCO_3^-$ transport mechanism affect the calcification process in different ways: (i) $HCO_3^-$ transport supplies the cell with substrate for the precipitation of $CaCO_3$ and (ii) the regulation of intracellular $HCO_3^-$ homeostasis is critical to buffer excess protons generated by the intravesicular precipitation of $CaCO_3$.

Apart from an endogenous generation of protons through biomineralization, pH regulatory mechanisms of sea urchin larvae have received considerable attention in the context of $CO_2$ driven sea water acidification (*Byrne et al., 2013*; *Stumpp et al., 2012*). The sea urchin larva has been extensively studied with respect to their potential for physiological acclimation and evolutionary adaptation to predicted near-future ocean acidification. These studies suggested that energy allocations and ion regulatory efforts are key processes that determine the resilience to reductions in seawater pH (*Pan et al., 2015*; *Pespeni et al., 2013*; *Stumpp et al., 2012*). These studies also indicated that despite moderate impacts at the organismal level, the compensatory reactions at the cellular level are substantial. Thus, a better mechanistic understanding for cellular processes affected by changes in seawater pH is essential to explain energy allocations in the sea urchin larva exposed to experimental ocean acidification.

Four Slc4 transporters were identified in the genome of the purple sea urchin, *Strongylocentrotus purpuratus* (*Tu et al., 2012*). To date, little is known regarding the function and tissue-specific localization of Slc4 transporters in the sea urchin larva. During the period of early skeleton formation in the sea urchin embryo (30–48 hr post fertilization), highest transcript abundance was detected for the *SpSlc4a10* gene compared to all other Slc4 transporters (*Tu et al., 2014*). This gene shares highest sequence identity with the mammalian Slc4a10 (NBCn2) gene that encodes an electroneutral sodium-dependnet $Cl^-$/$HCO_3^-$ exchanger with $Cl^-$ self-exchange activity (*Parker et al., 2008*). This prompted us to hypothesize that *SpSlc4a10* may be critically involved in $HCO_3^-$ transport during formation of the larval skeleton. Here, we test the role of *SpSlc4a10* in the maintenance of intracellular pH homeostasis and as a DIC concentration

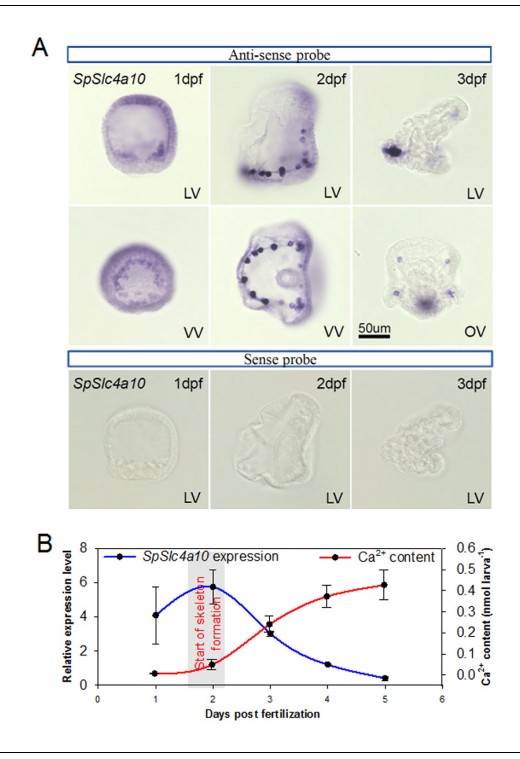

**Figure 1.** Expression pattern of the *SpSlc4a10* gene from blastula through pluteus larva in *Strongylocentrotus purpuratus*. (**A**) Localization of *SpSlc4a10* expression in the sea urchin larva along the larval development until 3 days post fertilization (dpf). Expression was detected in primary mesenchyme cells (PMCs) of the late blastula stage forming a ring around the blastopore. In the early pluteus larva *SpSlc4a10* expression is exclusively found in PMCs located at ends of the spicules. (**B**) *SpSlc4a10* expression levels and total calcium content along the early development of sea urchin larvae raised under control conditions. Bars represent mean ± SD; *n* = 3. dpf: days post fertilization; LV, lateral view; VV, vegetal view; OV, oral view.

DOI: https://doi.org/10.7554/eLife.36600.003

The following figure supplements are available for figure 1:

**Figure supplement 1.** Expression of *SpSlc4a10* in the syncytial cables of the PMCs.

DOI: https://doi.org/10.7554/eLife.36600.004

**Figure supplement 2.** Phylogeny of vertebrate and sea urchin Slc4 transporters.

DOI: https://doi.org/10.7554/eLife.36600.005

**Figure supplement 3.** Sequence alignment of the human NBCn2 (Slc4a10) and the sea urchin SpSlc4a10.

DOI: https://doi.org/10.7554/eLife.36600.006

mechanism in the calcifying PMCs of the sea urchin embryo. Since $pH_i$ regulation is intrinsically linked to precipitation of ACC, these mechanisms will fill an important knowledge gap regarding the fundamental principles of biomineralization in the sea urchin larva.

## Results

### Expression pattern of the sea urchin *SpSlc4a10* bicarbonate transporter

The *SpSlc4a10* has widespread expression in blastula embryos at 1 day post-fertilization (dpf), including the PMCs surrounding the blastopore (*Figure 1A*). In the late gastrula (prism stage, 2 dpf), *SpSlc4a10* is highly expressed in the PMCs during the formation of the syncycial cables. Besides a strong staining within the main cell bodies, *SpSlc4a10* is also expressed in the syncycial cables and filopodia of PMCs (*Figure 1—figure supplement 1*). In the pluteus larva (3 dpf) expression of *SpSlc4a10* was exclusively detected in PMCs located at the tips of the primary and secondary rods. Negative control using sense probes validate the specificity of the staining (*Figure 1A* lower panel). Among the four SLC4 transporters identified in the sea urchin genome (*SpSlc4a3*, *SpSlc4a2*, *SpSlc4a11* and *SpSlc4a10*) the PMC specific *SpSlc4a10* clusters within the clade of the Slc4a7-10, electroneutral $Na^+$-coupled $HCO_3^-$ transporters found in vertebrates (*Figure 1—figure supplement 2*). The amino acid sequence of the sea urchin SpSlc4a10 shares 44% similarity to the mammalian Slc4a10 (NBCn2) orthologue (*Figure 1—figure supplement 3*). During the first 5 days of development, *SpSlc4a10* expression peaks at 2 dpf, accompanied by the onset of larval $Ca^{2+}$ accumulation (*Figure 1C*). After this, expression levels decrease, while high $Ca^{2+}$ uptake rates are maintained up to 5 dpf.

### Phenotype of the *SpSlc4a10* morphant

Light microscopic analyses using polarized light (birefringence) demonstrated a disturbed formation of the larval skeleton in morphants compared to KCl injected control larvae (*Figure 2A*) or scramble MO control larvae (150 µM, *Figure 2—figure supplement 1*). The length of the primary rod is decreased in a dose-dependent manner with increasing *SpSlc4a10* morpholino (MO) concentrations (*Figure 2B*), while the tri-partite digestive tract develops normally. This reduction in primary rod length is accompanied by a decreased total $Ca^{2+}$ content in *SpSlc4a10* morphants (*Figure 2B*, inset). Besides the predominant expression of *SpSlc4a10* in PMCs, a $Na^+$/$K^+$-ATPase (*SpAtp1a3*) and a $Na^+$/$H^+$-exchanger (*SpSlc9a2*) isoform that are mainly expressed in stomach epithelial cells are also expressed in PMCs of late gastrula and pluteus larvae (*Figure 2C*). A four-fold up-regulation of *SpSlc4a10* expression levels in the morphants compared to control larvae at 3dpf demonstrates the specificity of the morpholino applied. In addition, *SpSlc9a2* was significantly downregulated in the morphants (*Figure 2D*). No differences were detected for the $Na^+$/$K^+$-ATPase coding gene *SpAtp1a3* between control larvae and the *SpSlc4a10* morphants (*Figure 2D*). In situ hybridization using the *SpSlc4a10* anti-sense probe demonstrates an irregular pattern of *SpSlc4a10* positive PMCs in the morphants compared to control larvae (*Figure 2E*).

### Immunohistology and westernblot analyses

Immunohistological analyses using a costom made antibody designed against the sea urchin SpSlc4a10 protein demonstrate strong immunoreactivity with primary mesenchyme cells and their syncytial cables (*Figure 3A*). High-magnification confocal microscopy indicates a localiazation of the protein in the plasma membrane as well as subcellular structures within the cytosol (*Figure 3B*). Negative controls by blocking the primary antibody with the immunizing peptide demonstrated no unspecific binding of the secondary antibody (*Figure 3C*). Westernblot analyses demonstrated positive immunoreactivity of our SpSlc4a10 antibody with a 135 KDa protein which is in the predicted size range of this protein (1183 amino acids; $\approx$ 130 KDa). Western-blot analyses demonstrate decreased SpSlc4a10 protein levels in MO-injected larvae when compared to control animals validating the knock-down by our SpSlc4a10 morpholino.

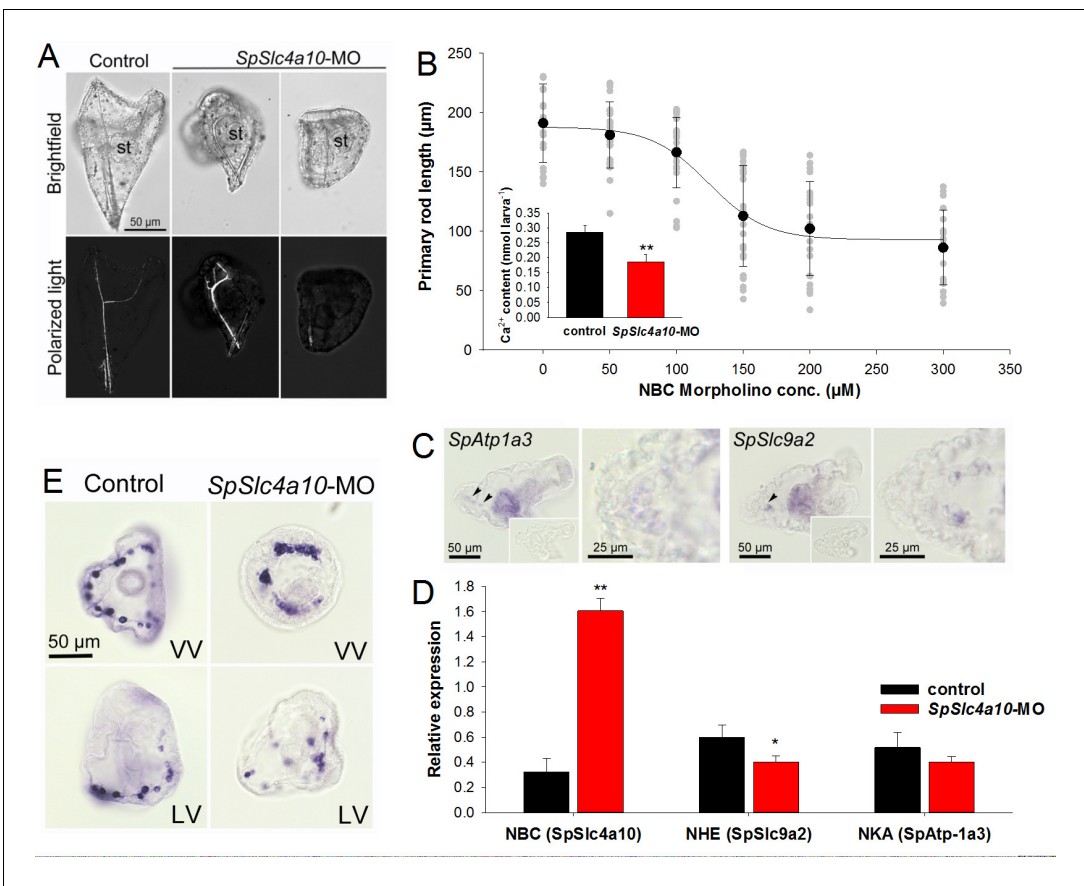

**Figure 2.** Morphological and molecular characterization of the *SpSlc4a10* morphant. (**A**) Light microscopic analyses with polarized light (birefringence) were used to detect deformations of the larval skeleton in *SpSlc4a10* morphants (4 dpf). The tri-partite digestive tract is normally developed in the *SpSlc4a10* morphants (st, stomach). (**B**) Length of the primary rod and total calcium content (inset) were used as indicators for reductions in calcification in 4 dpf morphants. Grey dots indicate individual measurements from experimental replicates (n = 4) (**C**) Expression of *SpAtp1a3* ($Na^+/K^+$-Atpase; NKA) and *SpSlc9a2* ($Na^+/H^+$-exchanger; NHE) in PMCs and stomach epithelial cells. (**D**) mRNA levels of *SpSlc4a10, SpAtp1a3 and SpSlc9a2* in control and *SpSlc4a10*-MO injected larvae. (**E**) Ring formation (2 dpf) of the *SpSlc4a10* expressing PMCs is disrupted in *SpSLC4a10* morphants (LV, lateral view; VV, vegetal view). Bars represent mean ± SD; *n* = 3–4 (*p<0.05; **p<0.01).
DOI: https://doi.org/10.7554/eLife.36600.007

The following figure supplement is available for figure 2:

**Figure supplement 1.** Comparison of KCL and scramble morpholino (MO) injected larvae.
DOI: https://doi.org/10.7554/eLife.36600.008

## Effect of $CO_2$-driven seawater acidification on $Ca^{2+}$ accumulation and *SpSlc4a10* expression

Under control conditions (pH 8.1), sea urchin embryos increase total $Ca^{2+}$ content at high rates of 1.75 nmol larva$^{-1}$ day$^{-1}$ during the formation of the larval skeleton (2–4 dpf) (*Figure 4A*). $CO_2$-induced reductions in seawater pH by 0.4 and 0.6 pH units resulted in a reduction in developmental rates (*Figure 4—figure supplement 1*) associated with a reduction in $Ca^{2+}$ accumulation (*Figure 4A*). Normalization of larval $Ca^{2+}$ content to bodylength (correction for developmental delay [*Stumpp et al., 2011*]) demonstrated no effect on the ability to accumulate $Ca^{2+}$ under acidified conditions (*Figure 4B*). Maintained calcification of larvae under acidified conditions is accompanied by an increased expression of *SpSlc4a10* at the onset of skeleton formation at 2 dpf (*Figure 4C*). This up-regulation of *SpSlc4a10* at the onset of skeleton formation is still evident when expression levels were normalized for body length correcting for the developmental delay (*Figure 4D*).

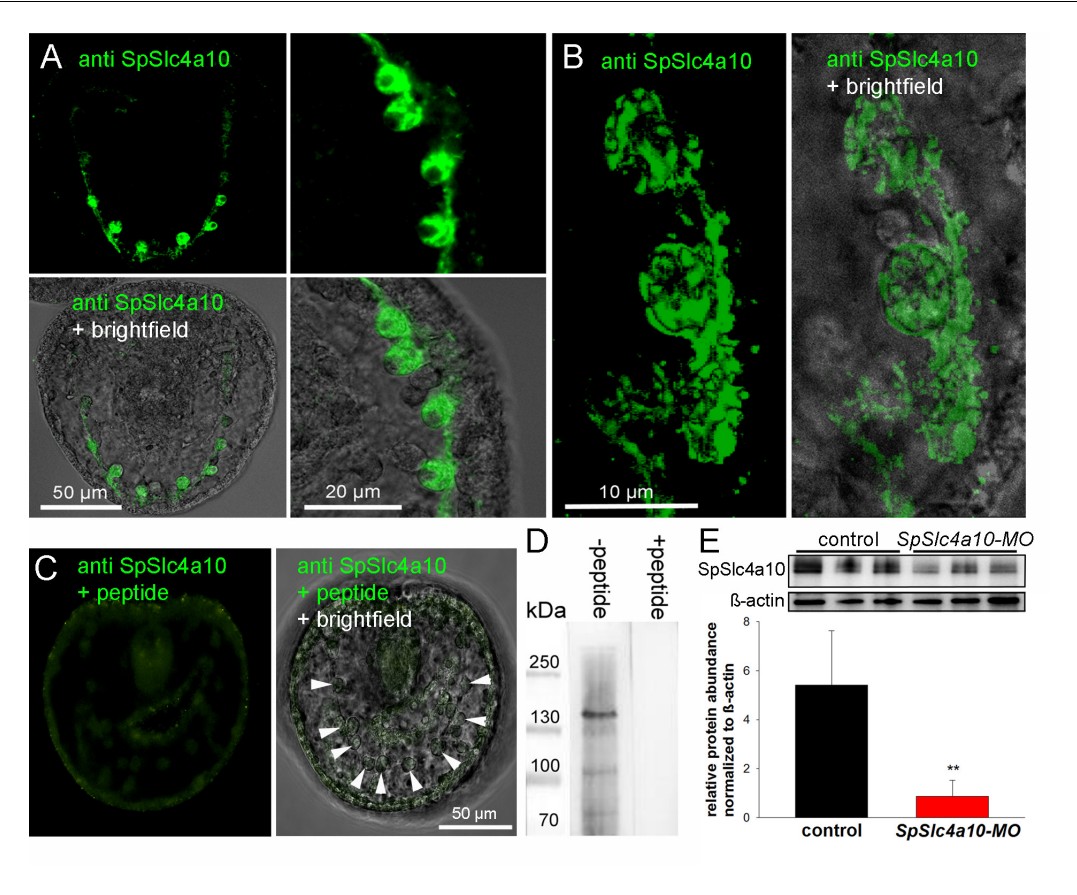

**Figure 3.** Localization of the SpSlc4a10 protein in PMCs and validation of the morpholino knock-down. (**A**) Immunohistological analyses using a custom made antibody designed against the sea urchin SpSlc4a10 protein demonstrating high concentrations of this protein in PMCs of late gastrula larvae (2 dpf). (**B**) High-magnification confocal microscopy showing the sub-cellular localization of the SpSlc4a10 protein. (**C**) Negative controls were performed by blocking the primary antibody with the immunizing peptide (PMCs indicated by arrowheads) (**D**) Westernblot analysis demonstrated specific immunoreactivity of the SpSlc4a10 antibody with a 135 KDa protein that disappeared in the peptide compensation assay. (**E**) Validation of the SpSlc4a10 knock-down by quantification of protein levels using western-blot analyses. Bars represent mean ±SD; $n$ = 3 (**p<0.01).
DOI: https://doi.org/10.7554/eLife.36600.009

## Intracellular pH regulation of PMCs

Intracellular pH ($pH_i$) regulatory abilities of PMCs were assessed by real-time ratiometric fluometry using the pH-sensitive dye BCECF-AM (*Figure 5A*). Exposure of PMCs from control animals to seawater containing 20 mM $NH_4Cl$ evoked an intracellular alkalosis by increasing $pH_i$ from 6.8 to 7.6. A slight compensation of the ammonia ($NH_3/NH_4^+$) induced alkalosis was observed during a period of 10 min. Removal of ammonia resulted in an intracellular acidosis ($pH_i$ = 6.1) that was efficiently compensated within 5 min (*Figure 5A,B*). *SpSlc4a10* morphants had a significantly decreased $pH_i$ (6.66 ± 0.17) compared to control larvae (6.87 ± 0.13) (*Figure 5B*; *Supplementary file 1*-Table S2), and a significantly reduced ability to recover from the ammonia-induced acidosis during washout of $NH_3/NH_4^+$. While control PMCs recover at a rate of 0.14 ± 0.05 pH units $min^{-1}$ that of the morphants was sigificantly decreased to 0.05 ± 0.02 pH units $min^{-1}$ (*Figure 5B,C*). Ratios were translated into pH values using the nigericin calibration method in combination with the depolarization of the membrane potential by high (150 mM) external $[K^+]$ (*Figure 5D*). Pharmacological experiments demonstrated that the recovery rate from an intracellular acidosis is sensitive to the anion exchange inhibitor 4,4'-Diisothiocyano-2,2'-stilbenedisulfonic acid (DIDS) in a dose-dependent manner (*Figure 5E*). Despite full inhibition of the DIDS-sensitive component of $pH_i$ regulation, PMCs remained capable of partly restoring $pH_i$ at lower rates in a similar fashion to that seen in the *SpSlc4a10* morphants (*Figure 5B,F*).

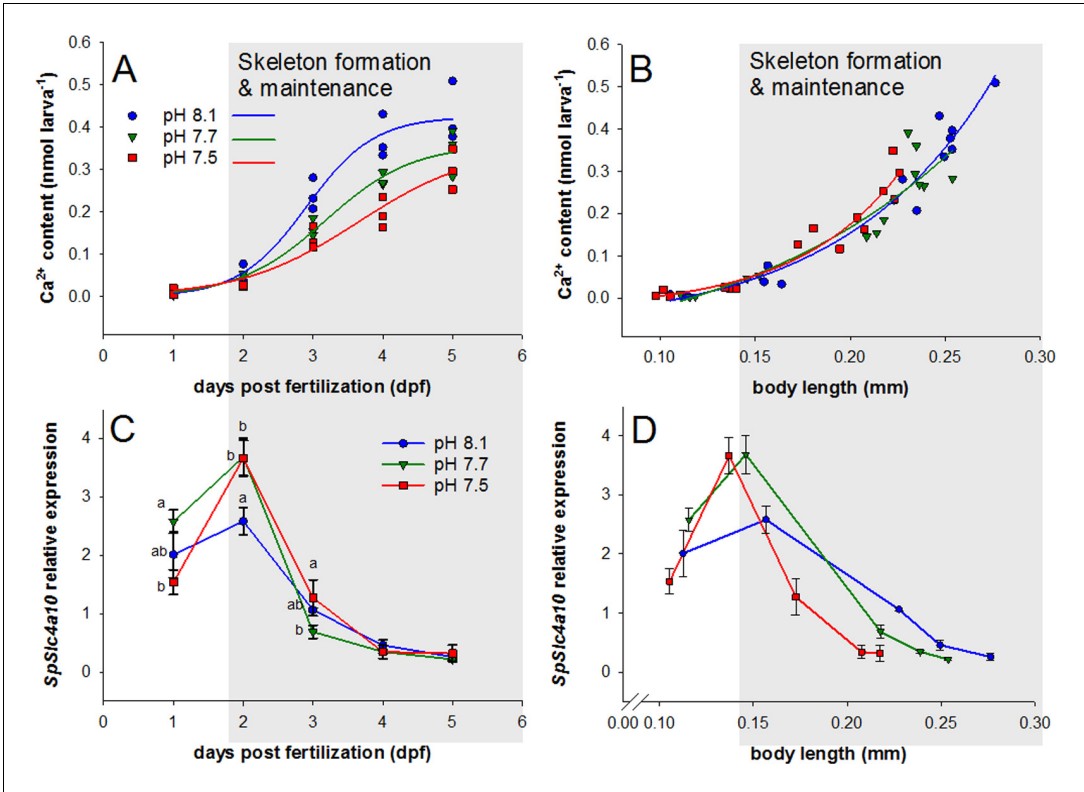

**Figure 4.** Development, calcium accumulation and expression of *SpSlc4a10* in sea urchin larvae raised under experimental ocean acidification. (**A**) Total Ca²⁺ content of larvae raised under three different pH conditions. (**B**) Larval Ca²⁺ content plotted as a function of body length to normalize for the developmental delay caused by acidified conditions (for morphometric analyses see Supplemental information *Figure 4—figure supplement 1*). (**C**) *SpSlc4a10* mRNA levels normalized to the housekeeping gene *SpZ12* during development under different pH conditions. (**D**) Expression pattterns for *SpSlc4a10* along the early development plotted as a function of body length. Different letters denote significant differences between treatments. Bars represent mean ± SD; *n* = 3.
DOI: https://doi.org/10.7554/eLife.36600.010

The following figure supplement is available for figure 4:

**Figure supplement 1.** Mortality and growth of sea urchin larvae raised under different pH treatments.
DOI: https://doi.org/10.7554/eLife.36600.011

## Calcein pulse-chase experiments

We tested for Ca²⁺ incorporation abilities in *SpSlc4a10* morphants (*Figure 6A,B*) and in larvae treated with the anion exchange inhibitor DIDS (*Figure 6C,D*) in comparison to control larvae. Late gastrula stage larvae were incubated in calcein for 3 hr and fluorescence intensities monitored using confocal microscopy. In addition to skeletal deformations, calcein incorporation was decreased in the *SpSlc4a10* morphants at the three different locations including posterior rod: PR (−43%), junction: JC (−31%) and anterior rod: AR (−35%). In the presence of 10 μM and 100 μM DIDS, fluorescence intensities in the three spicule sections, were also significantly decreased. Average values of different spicule sections (inset *Figure 6B*) demonstrated 40% and 50% reductions in spicule fluorescence intensity in the presence of 10 μM and 100 μM DIDS, respectively.

## Discussion

This study identified a SLC4-type bicarbonate transporter specifically expressed by primary mesenchyme cells (PMCs) of the sea urchin larva and demonstrated its role in the formation of the larval skeleton. In mammals, the SLC4 familiy can be categorized into three major clades of HCO₃⁻ and/or CO₃²⁻ transporters including Na⁺-driven Cl⁻/HCO₃⁻ exchangers, electrogenic Na⁺ coupled HCO₃⁻

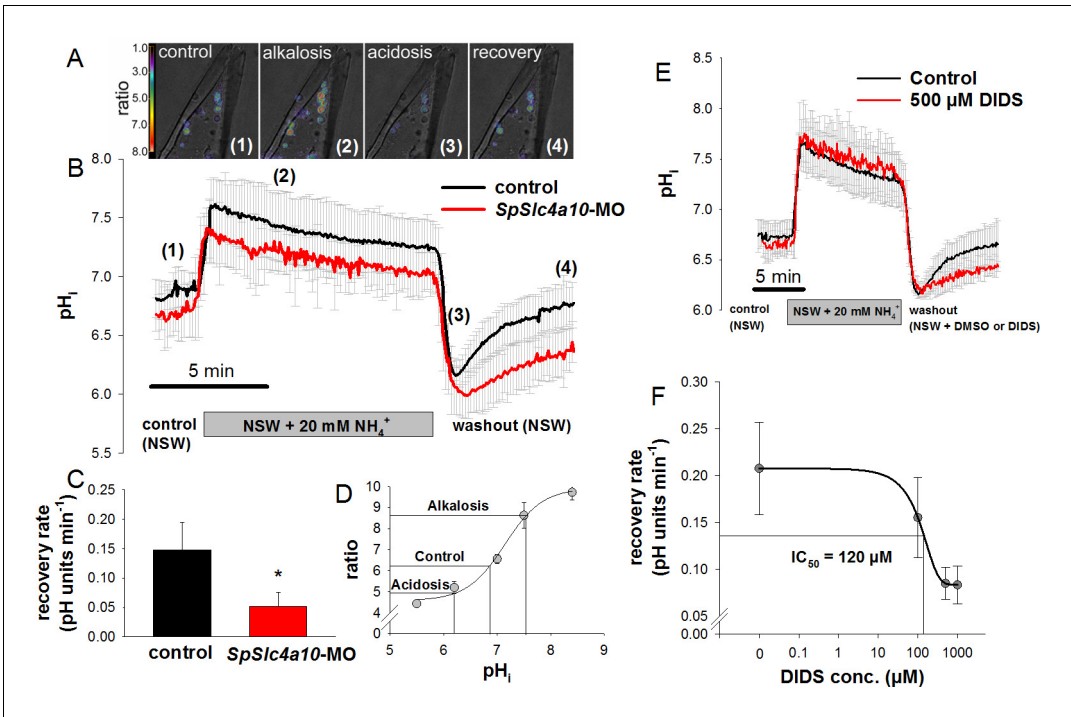

**Figure 5.** Intracellular pH regulatory abilities of primary mesenchyme cells. (**A**) Ratiometric fluorimetry in primary mesencyme cells (PMCs) using the pH sensitive dye BCECF-AM. False colour images superimposed on transmission images at time points 1, 2, 3, 4 as indicated in (**B**). (**B**) Summarized data from the control period (control (1)), after addition and removal of $NH_3/NH_4^+$ (alkalosis (2) and acidification (3); ammonium pulse), and during $pH_i$ recovery (4). (**C**) The recovery rate of the *SpSlc4a10* morphants was significantly reduced (see ***Supplementary file 1***-table S2 for summary of prameters measured). (**D**) Calibration curve of BCECF-AM in PMCs obtained at different pH levels in the presence of the ionophore nigericin and 150 mM $K^+$ allowing the translation of ratios to pH values. (**E**) Acid–base regulatory abilities of PMCs in the presence of 500 μM DIDS or only the vehicle (DMSO) as control. (**F**) The recovery rate from an intracellular acidosis is inhibited by DIDS in a dose-dependent manner with an $IC_{50}$ value of 120 μM. Bars represent mean ±SD; *p<0.05 ($n$ = 4–5 larvae with 3–5 cells measured per larvae).

DOI: https://doi.org/10.7554/eLife.36600.012

transporters and electroneutral $Na^+$ coupled $HCO_3^-$ transporters (***Romero et al., 2013***; ***Romero et al., 2004***). Our phylogenetic analysis based on deduced amino acid sequences suggest that *SpSlc4a10* has highest identity with the mammalian clade of electroneutral $Na^+$ coupled $HCO_3^-$-transporters, including $Na^+/HCO_3^-$ cotransporters (NBCn) and $Na^+$-driven $Cl^-/HCO_3^-$ exchangers (NCBE) that were demonstrated to control $pH_i$ in neurons (***Wang et al., 2000***) and enable the trans-cellular passage of $HCO_3^-$ across epithelial cells of the pancreatic duct and the renal proximal tubule (***Guo et al., 2017***; ***Ko et al., 2002***). The human NBCn2 encoded by *Slc4a10* has absolute requirements for $Na^+$ and $HCO_3^-$ and appears to be a $Na^+$-dependent $Cl^-/HCO_3^-$ exchanger that is blocked in the presence of 200 μM DIDS (***Parker et al., 2008***). Earlier studies demonstrated that $pH_i$ regulation in PMCs is also highly $Na^+$ and $HCO_3^-$-dependent thereby supporting the involvement of an $Na^+$-dependent $HCO_3^-$ transport mechanism in the regulation of PMC $pH_i$ (***Stumpp et al., 2012***). Together with findings of the present work demonstrating that $pH_i$ regulation in PMCs is DIDS sensitive ($IC_{50}$ = 120 μM) and that knock-down of *SpSlc4a10* reduces the ability to compensate an intracellular acidosis, there is strong evidence that *SpSlc4a10* plays a central role in controlling $pH_i$ of PMCs.

ACC is precipitated in intracellular vesicles which are subsequently exocytosed into the luminal space of the syncycial cable to form and maintain the larval skeleton (***Vidavsky et al., 2014***). Accordingly, protons generated during the formation of ACC must be removed from the vesicles into the cytosol to avoid an increasing intravesicular acidification leading to inhibition of $CaCO_3$ precipitation.

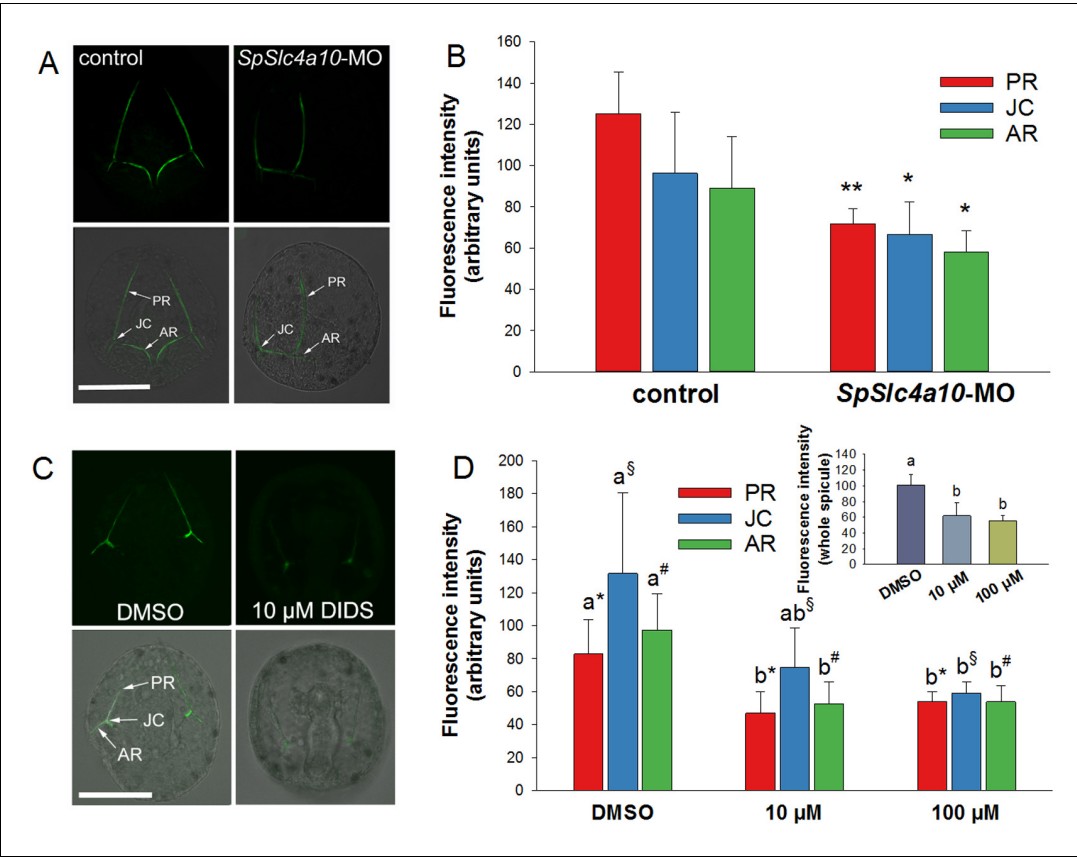

**Figure 6.** Calcein pulse–chase experiments to determine $Ca^{2+}$ incorporation into spicules during inhibition and knock-down of $HCO_3^-$ transport pathways. (A) Larvae of the late gastrula stage (2 dpf) were incubated in 160 μM calcein for 3 hr and confocal microscopy was used to determine fluorescence intensities in the spicules at different locations (posterior rod: PR, junction: JC, anterior rod: AR) after the calcein pulse. (B) Flourescence intensities are decreased in the morphants in all three locations. (C) Calcein incorporation in the presence of 10 μM DIDS or only the vehicle (DMSO) as control. (D) Fluorescence intensities, reflecting the amount of $Ca^{2+}$ precipitated into spicules during the calcein pulse, decreased in DIDS-treated larvae in a dose-dependent manner (inset: intensities for entire larval skeleton). Values are presented as mean ± SD; *n* = 4 (with 2–3 larvae per replicate experiment: 9–11 individuals). Different letters denote significant diffrences between treatments. Same symbols used in *Figure 5D* indicate groups that were compared by one-way ANOVA. *p<0.05; **p<0.01.
DOI: https://doi.org/10.7554/eLife.36600.013

This uptake of protons into the cytosol may cause disturbances in $pH_i$ homeostasis if not actively compensated by buffering and secretion of protons. Here, the PMCs of the sea urchin larva share many common features with mammalian osteoblasts where precipitation of the calcium phosphate mineral, hydroxyapatite, is initiated in vesicles (*Boonrungsiman et al., 2012*). These vesicles are exocytosed from osteoblasts to deliver mineral to the protein matrix of the bone (*Anderson, 2003*). During mineral precipitation large amounts of acid are produced that need to be removed in order to maintain the slightly alkaline conditions at the site of bone formation (*Brandao-Burch et al., 2005*; *Liu et al., 2011*). Thus, osteoblasts also possess a high $pH_i$ regulatory capacity during biomineralization that is achieved by a basolateral $Na^+/H^+$ exchanger 1 (NHE1) and NHE6 as well as $HCO_3^-/Cl^-$ exchange mechanisms by the anion exchanger 2 (AE2) (*Liu et al., 2011*; *Mobasheri et al., 1998*). The present work demonstrates that similar to mammalian osteoblasts, PMCs of the sea urchin larva also possess $pH_i$ regulatory mechanisms to promote biomineralization. Highest expression of the PMC-specific bicarbonate transporter *SpSlc4a10* is accompanied by the onset of skeleton formation in the sea urchin embryo and expression levels decrease in later stages that have lower calcification rates. The progressive down regulation of *SpSlc4a10* is accompanied by

a migration of *SpSlc4a10* expressing cells toward the tips of the spicules during further development.

Immunocytochemical analyses using a custom made antibody designed against the sea urchin SpSlc4a10 demonstrated high concentrations of this protein in PMCs. This suggests substantial $HCO_3^-$ transport capacities by PMCs that may be associated with $pH_i$ regulation and accumulation of $HCO_3^-$ to promote precipitation of $CaCO_3$. Besides a localization of SpSlc4a10 in the plasma membrane, large intracellular compartments demonstrated positive immunoreactivity as well. In mammalian skeletal tissues AE2 has been associated with the Golgi apparatus, where it is hypothesized to control pH within the Golgi complex (*Bronckers et al., 2009*; *Holappa et al., 2001*). By controlling pH inside the Golgi, AE2 is believed to mediate posttranslational modification of matrix proteins, their packaging, transfer to the outer plasma membrane and exocytosis (*Bronckers et al., 2009*; *Jansen et al., 2009*). A lack of this protein in skeletogenic cells is often associated with osteoporosis-like skeletal defects (*Jansen et al., 2009*). Accordingly, future studies will address the function of the SpSlc4a10 transporter in subcellular compartments and organelles of PMCs.

The disturbance of skeletogenesis in the *SpSlc4a10* morphants may be due to multiple factors, including (i) disturbance of PMC migration (ii) reductions in $HCO_3^-$ supply (iii) decreased control of $pH_i$. The phenotype of the *SpSlc4a10* morphants was characterized by decreased calcification abilities and abnormally formed spicules, lacking or having irregular branchings from the primary rods. Localization of *SpSlc4a10* positive PMCs in control and morphants indicated an abnormal migration of PMCs. Disturbance of migratory patterns and syncytium formation has been documented in several earlier studies using pharmacological approaches. For example loop diuretics, blocking the $Na^+/K^+/2Cl^-$ cotransporter (NKCC) inhibited the fusion and formation of cytoplasmic cords (*Basse et al., 2015*). Also the $H^+/K^+$-ATPase (HKA) inhibitor SCH28080 decreased calcification through impaired PMC fusion. However, while inhibition of HKA decreased $pH_i$ of PMCs, no changes in the migratory pattern were observed (*Schatzberg et al., 2015*). In the present study, the disruption of cell migration in the *SpSlc4a10* morphants may be explained by changes in $pH_i$ gradients that can be critical for cell polarization (*Martin et al., 2011*). It has been demonstrated that inhibition of NHE1, an important regulator of $pH_i$, caused $pH_i$ gradients to flatten or disappear. These gradients serve as an axis of movement in migrating cells and so their absence would likely affect migration (*Martin et al., 2011*). This hypothesis would be underlined by the down regulation of the $Na^+/H^+$-exchanger (*SpSlc9a2*) in the *SpSlc4a10* morphants. Although these are likely explanations for the observed migratory defects, intracellular pH disturbances were demonstrated to be associated with a wide range of cellular dysfunctions (e.g. *Kurkdjian and Guern, 1989*; *Madshus, 1988*; *Roos and Boron, 1981*) that may also systemically disturb PMC migration.

The present work provides further evidence for the requirement of bicarbonate transport pathways to regulate $pH_i$ that are likely the primary cause for reduced and abnormal skeleton formation (*Figure 7*). We propose that reductions in calcification observed for the *SpSlc4a10* morphants resulted from the decreased $pH_i$ regulatory ability of PMCs. The uncompensated proton load generated by the formation of ACC leads to an intracellular acidosis impairing futher precipitation of $CaCO_3$. Furthermore, since 40% of the dissolved inorganic carbon (DIC) that is incorporated into spicules derives from the seawater (*Sikes et al., 1981*), it can be suggested that *SpSlc4a10* may be also involved in a $HCO_3^-$ concentrating mechanism to promote intracellular calcification. Corroborating ealier studies that demonstrated reductions in $Ca^{45}$ uptake into spicules in the presence of the anion exchange inhibitor, DIDS (*Yasumasu et al., 1985*), the present work could show reduced incorporation of calcein in larvae treated with this compound. These reductions in calcification rates are accompanied by decreased $pH_i$ regulatory abilities under DIDS treatment. However, DIDS is a broad-band inhibitor for most SLC4 transporters, and has been demonstrated to potentially also inhibit the gastric-type $H^+/K^+$-ATPase in mammals (*Sugita et al., 1999*). Accordingly, it remains unresolved whether decreased $pH_i$ regulatory abilities under DIDS treatment can be attributed to an inhibition of $HCO_3^-$ or $H^+$ transport. Nonetheless, the knock-down of the $HCO_3^-$ transporter *SpSlc4a10* caused similar reductions in $pH_i$ regulatory capacities and calcein incorporation into spicules providing strong evidence for *SpSlc4a10* to be a key player controlling intracellular ACC formation in the sea urchin larva.

$CO_2$-perturbation experiments relevant for near-future ocean acidification scenarios demonstrated that sea urchin larvae are capable of maintaining calcification rates despite reductions in seawater pH. Reductions in seawater pH were demonstrated to negatively impact calcification in many

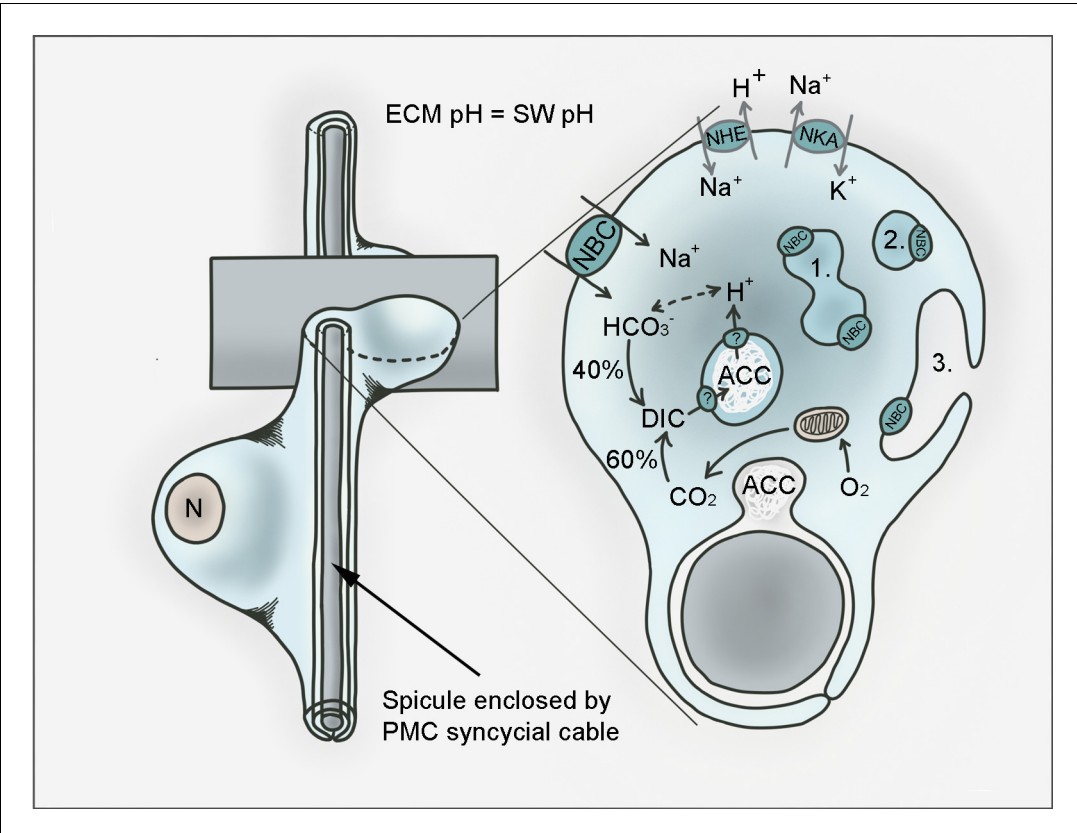

**Figure 7.** Hypothetical model for pH regulation and bicarbonate transport in PMCs of the sea urchin larva. PMCs form a syncytium within the extraxellular matrix (ECM) that has a pH the same as sea water (SW) in which the larva develops. Amorphous calcium carbonate (ACC) is precipitated in intracellular vesicles and exocytosed to the growing calcite spicule. Dissolved inorganic carbon (DIC) is provided through endogenous (i.e. respiratory $CO_2$) as well as exogenous (from the sea water) sources. *SpSlc4a10* (NBC) is proposed to mediate the import of bicarbonate from the seawater and to buffer protons generated during the precipitation of $CaCO_3$. Protons are exported from the vesicles through so far unknown pathways. Protons accumulating in the cytoplasm are potentially exported by the $Na^+/H^+$-exchanger (NHE) *SpSlc9a2*. Both secondary active transporters, NBC and NHE are driven by the $Na^+/K^+$-ATPase (NKA; *SpAtp1a3*) that is highly expressed by PMCs. In addition to its localization in the plasma membrane, NBC is associated with intracellular compartments including vesicles (1), vesicular networks (2) and vesicles fusing with the plasma membrane (3). n; nucleus.
DOI: https://doi.org/10.7554/eLife.36600.014

marine organisms due to a reduced availability of calcification substrates (e.g. $HCO_3^-$ and $CO_3^{2-}$) accompanied by enhanced dissolution of $CaCO_3$ (*Kroeker et al., 2013*). To maintain calcification rates under acidified conditions PMCs must allocate energy to increase $HCO_3^-$ / $CO_3^{2-}$ transport to the site of calcification and to export protons against steeper $H^+$ gradients (*Stumpp et al., 2012*). While the concept of energy allocation toward compensatory processes (i.e. pH regulation) is widely accepted, the identification of key acid-base transporters remains largely unknown. This work has identified *SpSlc4a10* as a key candidate gene that is critical for calcification under acidified conditions in the sea urchin larva.

## Conclusion

Here, we show that bicarbonate transport by a Slc4 family transporter is required for $pH_i$ regulation in sea urchin PMCs critical for the normal development of the larval skeleton. Interestingly, besides their importance in bone formation of mammals and vertebrates, the occurrence of specific bicarbonate transporters has been suggested to be a key step for the evolution of biomineralization in basal metazoans (*Zoccola et al., 2015*). A putative electroneutral $Na^+$-independent $Cl^-/HCO_3^-$

cotransporter SLC4γ has been exclusively associated with calcifying tissues of scleratinian corals that is lacking in other non-calcifying cnidarians. Based on the findings of the present work, future studies will use the sea urchin larva as a model to identify other PMC-specific acid-base transporters (e.g. $Na^+/H^+$-exchangers) that may be critically involved in the calcification process. Here, special attention will also be dedicated to the ability of PMCs to re-absorb the larval skeleton close to metamorphosis, which is also likely associated with pH modulations in the PMC syncycium. Since pH regulation is ultimately linked to the biological precipitation of $CaCO_3$, the present work provides a first step toward characterizing ion-regulatory mechanisms in PMCs critical for biomineralization the sea urchin embryo. This knowledge can then serve as a basis to identify conserved mechanisms of biomineralization in marine species and their potential for physiological buffering in times of rapid environmental change.

## Materials and methods

### Larval cultures and $CO_2$ perturbation experiments

Adult purple sea urchins (*Stongylocentrotus purpuratus*) were collected from the California coast (Kerckhoff marine Laboratory, California Institute of Technology), transferred to the Helmholtz Centre for Ocean Research Kiel (GEOMAR), and maintained in a re-cirulating natural seawater system at 11°C. Animals were fed with *Laminaria* sp. Spawning of males and females was induced by shaking and larval cultures were maintained as previously described (*Stumpp et al., 2013*; *Stumpp et al., 2015*). From 3 days post fertilization (dpf) larvae were fed daily with *Rhodomonas* spp. at a concentration of 10,000 cells $mL^{-1}$.

pH perturbation experiments were performed as previously described (*Stumpp et al., 2012*). Briefly, larval cultures with three replicates per pH treatment were continuously aerated with $CO_2$-enriched air providing constant $pCO_2$ levels of 400 μatm (pH 8.1), 1250 μatm (pH 7.7) and 4000 μatm (pH 7.5). Seawater physicochemical parameters including salinity, pH and temperature were monitored on a daily basis and samples for the determination of total dissolved inorganic carbon ($C_T$) were collected at two time points. The seawater carbonate system was calculated from $pH_{NBS}$ and $C_T$ using CO2sys as previously described (*Stumpp et al., 2013*) (see *Supplementary file 1*-Table S3). Larval densities were determined and paraformaldehyde (4% in filtere seawater) fixed samples were collected every day to monitor mortality and morphometry of larvae along the experimental period of 5 days.

### Phylogenetic analysis and molecular cloning

The amino acid sequences for phylogenetic analyses were collected from the Ensembl (www.ensembl.org) and EchinoBase (www.echinobase.org) databases. Collected sequences were submitted to the online tool MAFFT (www.ebi.ac.uk/Tools/msa/mafft/) for alignment and phylogenetic analysis. The results of the phylogenetic analysis were plotted using the phylogenetic tree generation tool provided on the iTOL website (itol.embl.de). For molecular cloning, the transcript sequences of sea urchin *SpSlc4a10*, *SpSlc9a2* and *SpATP1a3* were PCR amplified using primers provided in *Supplementary file 1*-Table S4. The amplicon sequence was then cloned into pGEM-Teasy (Promega) cloning vector and used to synthesize the RNA probes for in situ hybridization.

### Morpholino injection

Micro injections were performed according to the protocol provided in *Warner and McClay, 2014*. The gene-specific morpholino-substituted antisense oligonucleotides (MO) 5'-GTTCAAGTTGTTTC TCAGTTCTCGT-3′ complementary to the start codon region of the sea urchin *Slc4a10* gene as well as the srambled morpholino were obtained from Gene Tools (Oregon). The MO was dissolved in 0.5 M KCl solution and was injected into the fertilized egg (one-cell stage) using a microinjection system (Picospritzer III, Parker) mounted on an inverse microscope (Zeiss ObserverD1) equipped with a cooling stage.

### RT-qPCR and whole mount in situ hybridization

RNA from control and MO injected larvae was isolated by using the Direct-zol RNA MicroPrep kit (Zymo Research). RNA samples were reverse transcribed by Super Scipt IV cDNA synthesis kit

(Invitrogen, Waltham, USA) for quantitative PCR (qPCR). Expression levels of the target genes were measured by qPCR using the 7500 Fast Real-Time PCR system (Applied Biosystems) and normalized to the housekeeping gene SpZ12. qPCR primers used in this study are listed in *Supplementary file 1*-Table S5. Whole mount in situ hybridization was performed as previously described (*Stumpp et al., 2015*).

### Immunofluorescence (IF) staining and westernblot (WB) analysis using a custom made antibody designed against the sea urchin SpSlc4a10 protein

IF and WB analysis was performed as previously described (*Stumpp et al., 2013*; *Stumpp et al., 2015*). Briefly, for IF larvae ware fixed in 4% paraformaldehyde dissolved in filtered seawater for 15 min and postfixed in ice-cold methanol for 1 hr. The polyclonal primary antibody was generated against a synthetic peptide corresponding to an internal region (LTRHRHHKQKKKKEPENKAYNKG RRKS) of the sea urchin SpSlc4a10 protein. The affinity chromatography purified antibody was diluted 1:250 and samples were incubated over night at 4°C. After washing, samples were incubated in the secondary antibody for 1 hr, and pictures were taken on a confocal microscope (Axiovert 200M, Zeiss, Germany).

For WB, 250 larvae were collected, weighted and extracted by gentle pipetting in 1:10 wt/vol of Lämmli loading buffer. Proteins were fractionated by SDS PAGE on 6.5% polyacrylamide gels, and transferred to nitrocellulose membranes (Bio-Rad), using a tank blotting system (Bio-Rad). Blots were preincubated for 1 hr at room temperature in TBS-Tween buffer containing 5% (wt/vol) bovine serum albumin. Blots were incubated at 4°C overnight in a 1:7500 dilution of the primary antibody (see previous section). After washing with TBS-T, the blots were incubated for 1 hr with horseradish peroxidase-conjugated goat anti-rabbit IgG antibody (Santa Cruz Biotechnology, Santa Cruz, CA) diluted 1:14,000 in TBS-T. Protein signals were visualized with the ECL Western Blotting Detection Reagents (GE Healthcare, Munich, Germany) and photographed with a Gel Doc 2000 system equipped with a CCD camera (BioRad). For the peptide compensation assays used in whole mount IF and WB analysis, the primary antibody was pre-absorbed with the immunization peptide at a concentration of 0.1 mg/ml for 12 hr at 4°C.

Intracellular pH ($pH_i$) measurements and ammonia pulse technique $pH_i$ determinations and ammonia pulse experiments were conducted as previously described (*Stumpp et al., 2012*). Control and morphant larvae were measured in an alternate mode and from each larva 4–5 cells were simultanously recorded and treated as one replicate (n = 1). Nigericin in combination with high external [$K^+$] (150 mM) was used to calibrate $pH_i$ with our detected emission ratio of BCECF in living PMCs as previously described (*Stumpp et al., 2012*). For the ammonia pulse experiments, all larvae were exposed to filtered sea water (NSW) followed by the 20 mM $NH_3/NH_4^+$ prepulse (NSW +20 mM $NH_4Cl$). Acidosis was induced by washout using NSW or by using NSW containing different concentrations of 4,4′-Diisothiocyanatostilbene-2,2′-disulfonic acid disodium salt (DIDS) or 0.1% DMSO as vehicle control, respectively.

### Calcium content determination

To investigate the calcium content of larvae subjected to different pH levels, 50 ml of water from the larval cultures with known larval density was collected and larvae were concentrated by centrifugation. For the determination of $Ca^{2+}$ content in control and *SpSlc4a10* morphants 250 larvae were picked using a pipette and centrifuged to remove the sea water. Larvae were quickly washed three times in distilled water and the supernatant was removed after the last centrifugation. Samples were dried for 24 hr at 37°C. The samples were dissolved in 10 M HCl for 15 min and samples were analyzed using a flame photometer (EFOX 5053, Eppendorf). Calcium concentrations were normalized per larva and expressed as nmol $larva^{-1}$.

### Calcein pulse-chase experiments

Late gastrula stage larvae (2dpf) were used for calcein labeling experiments. Control and MO injected larvae were bathed in filtered seawater (NSW) containing 160 µM calcein (Sigma) for 3 hr. For inhibitor experiments, larvae were additionally exposed to different DIDS concentrations (0, 10 and 100 µM) in the calcein solution. Vehicle controls contained 0.1% of DMSO. After 3 hr of

incubation, larvae were washed four times by centrifugation and replacement of the calcein containing NSW by fresh NSW with the respective DIDS or DMSO concentration. Larvae were further incubated for 4 hr to remove calcein caught in the extracellular matrix of the larvae. Life larvae were mounted to microscope slides using synthetic cotton filaments (30–50 μm diameter) as spacers and covered with a coverslip. 5 Z-stacks images with a section thickness of 10 μm were aquired for every larva using a confocal microscope (Zeiss LSM 510) and fluorescence intensities were determined using the associated Zeiss imaging software (ZEN).

## Statistical analyses

Data were tested for normality and homogeneity and significance levels were analyzed using paired t-tests. For the analyses of gene expression in $CO_2$-treated larvae as well as comparisons of calcein fluorescence intensities in morphants and DIDS-treated larvae one-way ANOVA followed by Holm-Sidak post-hoc test was used. Regression analyses were performed on log transformed data using linear models. ANCOVA followed by the Holm-Sidak method was performed to test for sigificant diffrences in $Ca^{2+}$ content under different $CO_2$ treatments. Statistical analyses were conducted using Sigma Stat 13 (Systat Software).

## Acknowledgements

We would like to thank M Thorndyke for constructive comments and discussions on this work. This project was funded a Cluster of Excellence 80 'The Future Ocean' (CP1409 and CP1519) grant to MH. The 'Future Ocean' is funded within the framework of the Excellence Initiative by the Deutsche Forschungsgemeinschaft (DFG) on behalf of the German federal and state governments.'

## Additional information

### Funding

| Funder | Grant reference number | Author |
|---|---|---|
| Deutsche Forschungsge-meinschaft | Cluster of Excellence CP1519 | Marian Y Hu |
| Deutsche Forschungsge-meinschaft | Cluster of Excellence CP1409 | Marian Y Hu |

The funders had no role in study design, data collection and interpretation, or the decision to submit the work for publication.

### Author contributions

Marian Y Hu, Conceptualization, Data curation, Formal analysis, Supervision, Funding acquisition, Investigation, Visualization, Methodology, Writing—original draft, Project administration, Writing—review and editing; Jia-Jiun Yan, Conceptualization, Investigation, Methodology, Writing—original draft; Inga Petersen, Data curation, Formal analysis, Validation, Methodology; Nina Himmerkus, Conceptualization, Data curation, Methodology, Writing—original draft, Writing—review and editing; Markus Bleich, Visualization, Methodology, Writing—original draft; Meike Stumpp, Conceptualization, Visualization, Methodology, Writing—original draft, Writing—review and editing

### Author ORCIDs

Marian Y Hu  http://orcid.org/0000-0002-8914-139X

### Decision letter and Author response

Decision letter https://doi.org/10.7554/eLife.36600.018
Author response https://doi.org/10.7554/eLife.36600.019

## Additional files

### Supplementary files

• Supplementary file 1. Table S1: List of species and gene accession numbers of sequences used for phylogenetic analysis of Slc4 family transporters. Table S2: pH regulatory parameters of primary mesenchyme cells in control and *SpSlc4a10* morphants along the ammonia pulse experiment. Intracellular buffercapacitiy (ß) was calculated using the equation: $\beta = \Delta[NH^{4+}]/\Delta pH_i$. Table S3: Seawater physico-chemical parameters monitored during the pH manipulation experiment. Parameters measured include Salinity, Temperature, pH (NBS scale), and total dissolved inorganic carbon ($C_T$). $pH_{NBS}$ and $C_T$ were used to calculate the carbonate system, including $pCO_2$, total alkalinity ($A_T$) and the saturation states for calcite ($\Omega_{Ca}$) and aragonite ($\Omega_{Ar}$). Table S4: List of primers used for molecular cloning. Table S5 List of primers used for qPCR.
DOI: https://doi.org/10.7554/eLife.36600.015

• Transparent reporting form
DOI: https://doi.org/10.7554/eLife.36600.016

### Data availability

All data generated or analysed during this study are included in the manuscript and supporting files.

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
