## [Decision Letter]

[Editors’ note: a previous version of this study was rejected after peer review, but the authors submitted for reconsideration. The first decision letter after peer review is shown below.]

Thank you for submitting your work entitled "A SLC4 family bicarbonate transporter regulates intracellular pH critical for biomineralization in the sea urchin embryo" for consideration by *eLife*. Your article has been reviewed by three peer reviewers, and the evaluation has been overseen by a Reviewing Editor and a Senior Editor. The following individuals involved in review of your submission have agreed to reveal their identity: Martin Tresguerres (Reviewer #1).

Our decision has been reached after consultation between the reviewers. Based on these discussions and the individual reviews below, we regret to inform you that your work will not be considered further for publication in *eLife*. However, should the authors address the concerns raised by the reviewers with respect to experimental controls and specificity, we would entertain a revised paper in the future. This would take the form of a new submission, linked to the previous one within the system and would be assessed by the same experts.

*Reviewer #1:*

This study attempted to characterize the involvement of the bicarbonate transporter *SpSlc4a10* in sea urchin larvae calcification. The hypothesis was that *SpSlc4a10* is preferentially expressed in primary mesenchyme cells (PMCs), which are responsible for calcification, where it alkalanizes intracellular pH (pHi) to promote calcification. The study also addressed the potential involvement of *SpSlc4a10* under ocean acidification (OA)- like conditions.

The topic is important and timely because information about molecular and cellular calcification mechanisms in marine organisms is scarce, and because this is essential for understanding and predicting potential effects of OA. The choice of experimental organism is excellent because sea urchins have very well characterized developmental biology and genome, and genetic manipulations are possible.

Unfortunately, this study has one major flaw in that the specificity and efficacy of *SpSlc4a10* morpholino knockdown was not validated.

Figure 2D shows a 4-fold upregulation of *SpSlc4a10* mRNA, however this is not a reliable validation. In addition, those larvae had significant NHE downregulation (which would also explain the results). It is likely that mRNA levels for other ion transporting proteins that were not examined were also up or downregulated. This makes it impossible to attribute the effects to *SpSlc4a10*.

Morpholinos should be validated in some of the following ways:

The gold standard would be to determine protein abundance using validated antibodies; *SpSlc4a10* protein level should be reduced. Antibodies against *SpSlc4a10* would have the additional advantage of determining subcellular localization -i.e. basolateral or apical membrane, or intracellular calcifying vesicles.

If making antibodies is not possible, a second morpholino against another region of the *SpSlc4a10* gene should be used.

Alternatively, CRISPR-Cas9 or siRNA could be used.

In any case, control "scrambled morpholino" should be used and those larvae must be used for at least some of the key experiments. Unfortunately, skeleton defects is a very common phenotype seen in stressed sea urchin larvae, so determining a specific effect on calcification requires very carefully controlled experiments.

Another issue is the use of DIDS. While it is true this drug inhibits bicarbonate transporters, it is very broad so it will inhibit all bicarbonate transporters (not just *SpSlc4a10*). Furthermore and even more worrying is that DIDS has multiple and significant off-target effects, which explains why cutting-edge biomedical research has stopped using DIDS at least a decade ago. For example, DIDS inhibits the H^+^/K^+^-ATPase at concentrations as low as 1μM (Sugita, et al. 1999).

One way to circumvent this problem would be to assess the effect of DIDS on cells from sea urchin larvae morphants (which should not be sensitive to DIDS).

Moreover, the effects of DMSO (the vehicle for DIDS) on pHi should be determined.

Unless these issues are resolved, I will not provide further feedback about the experiments or conclusions. I repeat this is a very interesting and important topic, and that the choice of experimental organism is excellent. However, the issues listed above must be addressed.

*Reviewer #2:*

This manuscript suggests that a member of the SLC4 HCO3 transporter family, *SpSlc4a10*, has an important role in pH regulatory and HCO3 concentrating mechanism is sea urchin.

This is a very interesting and important information relating to the biomineralization mechanism in animals. This is a very elegant and well design experiment, the authors studied each relevant aspect that HCO3 transporter can affect.

*Reviewer #3:*

This manuscript addressed the role of bicarbonate transport in developing sea urchins, in particular on biomineralization. The manuscript is well written and easy to follow. The manuscript supports a key role of the bicarbonate transporter *slc4a10* in pH regulation and biomineralization processes in developing sea urchin. The only real problem with the manuscript is a logical gap as to why the research has chosen to focus on *slc4a10* as opposed to any other bicarbonate transporter of sea urchins? The data do support a central, but not unique, role of *slc4a10* in the processes mentioned above.

1) Fig, 1B is too small to be of value. It should be greatly increased in size and included as a figure supplement.

2) Figure 1C needs an X-axis label (presumably time).

3) Subsection “Expression pattern of the sea urchin *SpSlc4a10* bicarbonate transporter”, first sentence. The term "ubiquitous" may be too strong. Perhaps "widespread" would be better supported by the data.

4) The manuscript does not provide sufficient text rationalizing the focus on *slc4a10*, to the exclusion of other bicarbonate transporters in sea urchin. Please explain why the analysis was confined to this transporter.

5) Figure 2 legend and elsewhere: Ideally legends should explain the experiment, but not interpret the results. The phrase "demonstrate developmental defects in the formation of the larval skeleton in the pluteus larva (4 dpf)" clearly belongs in the Results section, not legend. Same issue arises in panel D and E legend. Please revise this and other legends accordingly.

6) The established convention in the SLC4 field is that NBC refers to "sodium bicarbonate co-transporter". Family member SLC4A10 is an electroneutral transporter called NBCn2, as the second recognized electroneutral sodium bicarbonate cotransporter. If the sea urchin gene product studied here is orthologous to mammalian SLC4A10, it may very well share the same transport function. If so, that function is electroneutral sodium dependent Cl^-^/HCO_3_^-^ exchange. This point should form part of the introduction. Moreover, in the manuscript where the term "NBC" is used to described *slc4a10*, NBCn2 might be preferable.

The manuscript needs to provide the evidence in the form of amino acid sequence alignments or literature citations that identifies the gene under investigation as orthologous to SLC4A10.

7) The third paragraph of the Discussion would be enhanced by citing literature discussing the myriad effects of pH on cellular function. So powerful is the effect of pH that it is difficult to assess the mechanism by which it acts.

---

## [Author Response]

[Editors’ note: the author responses to the first round of peer review follow.]

Reviewer #1:

This study attempted to characterize the involvement of the bicarbonate transporter SpSlc4a10 in sea urchin larvae calcification. The hypothesis was that SpSlc4a10 is preferentially expressed in primary mesenchyme cells (PMCs), which are responsible for calcification, where it alkalanizes intracellular pH (pHi) to promote calcification. The study also addressed the potential involvement of SpSlc4a10 under ocean acidification (OA)- like conditions.

The topic is important and timely because information about molecular and cellular calcification mechanisms in marine organisms is scarce, and because this is essential for understanding and predicting potential effects of OA. The choice of experimental organism is excellent because sea urchins have very well characterized developmental biology and genome, and genetic manipulations are possible.

We would like to thank reviewer #1 for a generally positive but also critical review on our manuscript. We spent a lot of time and efforts to carefully address and resolve all issues raised by reviewer #1, which has significantly increased the quality and scientific value of this work.

We were able to generate and validate an antibody raised against the sea urchin SpSlc4a10 protein. Using this antibody we validated our MO-knock-down by western blot analysis and could confirm our in situ hybridization experiments demonstrating that also the protein of SpSlc4a10 is exclusively located in PMCs. We performed additional experiments using scrambled morpholinos to demonstrate that skeletal defects can be specifically attributed to the SpSlc4a10 MO. In addition, in the context of another project, we are currently working with several other MOs targeting proteins expressed in the endoderm of the sea urchin larva. Using these MOs we found no skeletal defects making us very confident that skeletal deformations and reductions in calcification seen in this study can be attributed to the specificity of our SpSlc4a10 MO. We also explained the rationale for choosing DIDS in our pharmacological experiments and the implications of these observations for the research community.

Unfortunately, this study has one major flaw in that the specificity and efficacy of SpSlc4a10 morpholino knockdown was not validated.Figure 2D shows a 4-fold upregulation of SpSlc4a10 mRNA, however this is not a reliable validation. In addition, those larvae had significant NHE downregulation (which would also explain the results). It is likely that mRNA levels for other ion transporting proteins that were not examined were also up or downregulated. This makes it impossible to attribute the effects to SpSlc4a10.Morpholinos should be validated in some of the following ways:The gold standard would be to determine protein abundance using validated antibodies; SpSlc4a10 protein level should be reduced. Antibodies against SpSlc4a10 would have the additional advantage of determining subcellular localization -i.e. basolateral or apical membrane, or intracellular calcifying vesicles.If making antibodies is not possible, a second morpholino against another region of the SpSlc4a10 gene should be used.Alternatively, CRISPR-Cas9 or siRNA could be used.

We agree with reviewer #1 that a compensatory expression, by increasing *SpSlc4a10* mRNA levels in morphants, merely represents a weak validation for the specificity of our knock-down. According to the reviewer´s suggestion we now included additional experiments to validate the specificity of our *SpSlc4a10* knock-down by demonstrating that protein levels are decreased in *SpSlc4a10* morphants. We generated a polyclonal antibody designed against AA205-231 (LTRHRHHKQKKKKEPENKAYNKGRRKS) of the *SpSlc4a10* protein of the purple sea urchin (*Strongylocentrotus purpurtus*). Western blot analyses demonstrate specific immunoreactivity of our *SpScl4a10* antibody with a 130 KDa protein which is in the predicted size range, and which is similar to the molecular weight of the mammalian *Slc4a10* orthologue. *SpSlc4a10* protein levels are significantly reduced by 70% (normalized to ß-actin) in morphants compared to control larvae. Immunocytological analyses demonstrate high concentration of this protein in primary mesenchyme cells and associated filopodia. The protein is localized in cellular membranes as well as in the cytosol (vesicles). We added this data as a new figure (Figure 3) to the revised manuscript and added the respective paragraphs to the manuscript describing and discussing these new findings.

In any case, control "scrambled morpholino" should be used and those larvae must be used for at least some of the key experiments. Unfortunately, skeleton defects is a very common phenotype seen in stressed sea urchin larvae, so determining a specific effect on calcification requires very carefully controlled experiments.

We used “scramble morpholino” at the same concentration as we applied our *SpSlc4a10* MO and could not detect any morphological defects during larval development. We added a new supplemental figure to this work comparing primary rod-length in KCL and “scramble-MO” injected larvae (Figure 2—figure supplement 1). Furthermore, using other MOs (at the same concentration) targeting endodermal genes we did not find skeletal defects underlining the specificity of skeletal deformations in our *SpSlc4a10* morphants.

Another issue is the use of DIDS. While it is true this drug inhibits bicarbonate transporters, it is very broad so it will inhibit all bicarbonate transporters (not just SpSlc4a10). Furthermore and even more worrying is that DIDS has multiple and significant off-target effects, which explains why cutting-edge biomedical research has stopped using DIDS at least a decade ago. For example, DIDS inhibits the H^+^/K^+^-ATPase at concentrations as low as 1μMSugita, et al. (1999).One way to circumvent this problem would be to assess the effect of DIDS on cells from sea urchin larvae morphants (which should not be sensitive to DIDS).Moreover, the effects of DMSO (the vehicle for DIDS) on pHi should be determined.

We agree that DIDS is a rather broad-band compound inhibiting most Slc4 transporters and potentially also other transporters relevant for pH homeostasis. There are two main purposes why we used DIDS in addition to our specific knock-down of *Slc4a10*. First, this inhibitor has been used in studies addressing the formation of the larval sea urchin skeleton (Yasumasu et al. 1986 Exp Cell Res 159:80-90). These studies demonstrated skeletal deformations and the authors concluded the involvement of HCO_3_^-^ transport in the biomineralization process with the underlying mechanisms remaining unresolved. In this way the present work helps demonstrating that DIDS affects pHi regulatory abilities that are likely responsible for skeletal defects in the sea urchin larva. Second, based on our previous studies demonstrating that pH regulation in PMCs is HCO_3_^-^ and Na^+^ dependent, we used DIDS to provide a complete chain of evidences (Na+, HCO_3_^-^ manipulations, pharmacology and morpholino knock-down of *SpSlc4a10*) demonstrating the importance of HCO_3_^-^ transport in pHi regulation.

In addition we added a paragraph discussing potential off-target effects of DIDS mentioned by reviewer #1: “These reductions in calcification rates are accompanied by decreased pH_i_ regulatory abilities under DIDS treatment. […] Accordingly, it remains unresolved whether decreased pH_i_ regulatory abilities under DIDS treatment can be attributed to an inhibition of HCO_3_^-^ or H^+^ transport.”

In our controls we always added the vehicle (DMSO) at the same concentration as in the ones with DIDS. We have this information in the Materials and methods section as well as in the respective figure legends.

Unless these issues are resolved, I will not provide further feedback about the experiments or conclusions. I repeat this is a very interesting and important topic, and that the choice of experimental organism is excellent. However, the issues listed above must be addressed.

We appreciate that reviewer #1 sees the importance and timeliness of this work and hope he is satisfied with the revised version now including the requested experiments.

Reviewer #3:This manuscript addressed the role of bicarbonate transport in developing sea urchins, in particular on biomineralization. The manuscript is well written and easy to follow. The manuscript supports a key role of the bicarbonate transporter slc4a10 in pH regulation and biomineralization processes in developing sea urchin. The only real problem with the manuscript is a logical gap as to why the research has chosen to focus on slc4a10 as opposed to any other bicarbonate transporter of sea urchins? The data do support a central, but not unique, role of slc4a10 in the processes mentioned above.

We are grateful for the positive and constructive comments made by reviewer #3 that significantly improved the quality of our manuscript. We addressed and accepted all concerns raised and corrected them in the revised version of our manuscript. In particular, we explained our rationale for choosing *SpSlc4a10* out of the six Slc4 transporters annotated in the sea urchin genome (see comments below). Furthermore we corrected all legends in that we now focus on explaining the experiments rather than describing the data. We also separated the Introduction and Discussion into shorter focused units and added a paragraph introducing the function of the mammalian NBCn2. We hope reviewer #3 is happy with the revised manuscript and we would like to once again thank him/her for constructive criticism.

1) Fig, 1B is too small to be of value. It should be greatly increased in size and included as a figure supplement.

We enlarged Figure A2 and moved it to the supplemental material part.

2) Figure 1C needs an X-axis label (presumably time).

We added the axis label (days post fertilization).

3) Subsection “Expression pattern of the sea urchin SpSlc4a10 bicarbonate transporter”, first sentence. The term "ubiquitous" may be too strong. Perhaps "widespread" would be better supported by the data.

We agree and replaced ubiquitous with widespread: “The *SpSlc4a10* has widespread expression in blastula embryos.”

4) The manuscript does not provide sufficient text rationalizing the focus on slc4a10, to the exclusion of other bicarbonate transporters in sea urchin. Please explain why the analysis was confined to this transporter.

We agree with the reviewer’s comment and added information as to why we specifically addressed the role of *SpSlc4a10* in this study. A developmental transcriptome (Tu, Cameron and Davidson, 2014) indicates that the *SpSlc4a10* gene is highly expressed in the early sea urchin embryo during skeleton formation. We added the following paragraph to the Introduction: “Six Slc4 transporters including four putative anion exchangers and two sodium-bicarbonate co-transporters were identified in the genome of the purple sea urchin (Tu et al., 2012). […] This prompted us to hypothesize that *SpSlc4a10* may be critically involved in HCO_3_^-^ transport during formation of the larval skeleton.”

5) Figure 2 legend and elsewhere: Ideally legends should explain the experiment, but not interpret the results. The phrase "demonstrate developmental defects in the formation of the larval skeleton in the pluteus larva (4 dpf)" clearly belongs in the Results section, not legend. Same issue arises in panel D and E legend. Please revise this and other legends accordingly.

We agree with the concern raised by reviewer #3 and carefully corrected and rephrased all respective figure legends.

6) The established convention in the SLC4 field is that NBC refers to "sodium bicarbonate co-transporter". Family member SLC4A10 is an electroneutral transporter called NBCn2, as the second recognized electroneutral sodium bicarbonate cotransporter. If the sea urchin gene product studied here is orthologous to mammalian SLC4A10, it may very well share the same transport function. If so, that function is electroneutral sodium dependent Cl^-^/HCO_3_^-^ exchange. This point should form part of the introduction. Moreover, in the manuscript where the term "NBC" is used to described slc4a10, NBCn2 might be preferable.The manuscript needs to provide the evidence in the form of amino acid sequence alignments or literature citations that identifies the gene under investigation as orthologous to SLC4A10.

We added information regarding the function of the mammalian NBCn2 to the introduction. We carefully checked our text and replaced *Slc4a10* by NBCn2 where necessary. We also added an amino acid sequence alignment of the human NBCn2 and the sea urchin *SpSlc4a10* to the supplemental part (Figure 1—figure supplement 3).

7) The third paragraph of the Discussion would be enhanced by citing literature discussing the myriad effects of pH on cellular function. So powerful is the effect of pH that it is difficult to assess the mechanism by which it acts.

We added a sentence to this paragraph referring to many cellular defects associated with intracellular pH disturbances: “However, intracellular pH disturbances were demonstrated to be associated with a wide range of cellular dysfunctions (e.g. Kurdjian and Guern, 1989; Madshus, 1988; Roos and Boron, 1981) that may contribute to the disturbed PMC migration observed in this study.”